# ALIA: An LLM for Industrial Assets using Synthetic Data

## Abstract

With the emergence of agentic workflow development using Large Language Models (LLMs) for industrial applications, there is a growing need for small language models to possess domain-specific knowledge. In many existing approaches, reference materials such as books are used as a source of knowledge. This paper presents a novel approach to finetune a base LLM model in a continued pre-training fashion for the industrial asset domain, leveraging knowledge documented in a tabular structure to generate synthetic knowledge documents and a vast amount of question-answer pairs using an entity and relationship-driven approach. Ultimately, this approach enables the fine-tuning of a small LLM (such as LLAMA 3.1-8B) for evaluating the performance enhancement it brings. We tested the base model and enhanced model on Industry4-FMSR MCQA datasets with 2600+ samples and obtained around 4% improvement overall. Our experimental results confirm the validity of our approach to generate the more synthetic data for a knowledge infusion task.

## 1 Introduction

The emergence of agentic workflows using Large Language Models (LLMs) for industrial applications has created a pressing need for smaller language models to possess domain-specific knowledge Chen et al. (2023), Liévin et al. (2023), Wu et al. (2023). While recent advancements in language models have demonstrated impressive fluency, they are not without flaws; these models can generate false statements, leading to errors in task execution Lin et al. (2022), Zhang et al. (2024b), Nori et al. (2023). In our recent benchmark study evaluating the truthfulness of LLMs in the context of industrial assets, we found that frontier model such as ChatGPT 4o was truthful approximately 65% of the time, on the other hand the performance of smaller models was a significant concern (around 40%). This discrepancy can adversely affect an agent's decision-making process, prompting us to address the challenge of improving their reliability. Many use cases, such as work order classification Stewart et al. (2023), require a solid understanding of the domain to function effectively.

Building an agentic workflow represents an important area of research that will enhance the adoption of LLMs Yao et al. (2022), Aksitov et al. (2023), Wang et al. (2024). In such frameworks, agents interact with their environments using natural language and express their reasoning similarly Yan et al. (2023). This underscores the necessity for agents to think like experts rather than merely functioning as token generators. Achieving this domain-specific reasoning involves a knowledge distillation process. For instance, an agent might internally contemplate, "I need to retrieve the failure mode", or "Can I detect a failure using sensor data?"

The majority of knowledge distillation processes begin with documents or existing knowledge and utilize a teacher model to synthetically generate numerous question-answer pairs, transferring this knowledge to smaller models Wang et al. (2022), Sudalairaj et al. (2024). Knowledge is often deeply encoded in various formats, including documents, tables, images, and time series data. However, extracting knowledge from structured forms to initiate synthetic data generation can be challenging.

This paper focuses on harnessing structured information available in a specific domain to systematically navigate the knowledge infusion process. In our exploration, we observed that many of our ISO documents contained critical knowledge encoded as tables. For example, we found valuable insights linking failure modes to sensor analytics and component mappings.

Applications such as failure prediction, anomaly detection, and work order classification rely heavily on extensive knowledge about assets and their processes. Building a specialized LLM for industrial applications is therefore desirable, but harvesting knowledge from trustworthy sources remains a challenge. ISO documents and established standards serve as reliable resources in this context. Our approach involves "crawling" the internal knowledge base of a language model with this limited knowledge. We developed an Industrial-FMSR-MCQA system and tested it on the Llama model Touvron et al. (2023) with varying parameters. Benchmark results demonstrated significant performance improvements using a model fine-tuned with synthetic data. This raises the question: can we leverage a larger model to extract information and import it into a smaller model's knowledge base? By devising a scheme to extract useful knowledge from a limited yet factual structured dataset, we aim to enhance smaller LLMs for specialized industrial tasks. Our key contributions are:

- We propose leveraging fine-grained information from structured tables in ISO standards to enhance model training. By extracting specific attributes related to equipment failure modes, we create enriched feature representations that improve the model's ability to generalize and accurately reason about new, unseen data during evaluation.

- We introduce a novel two-step iterative approach that integrates a Knowledge Graph (KG) with a Large Language Model (LLM) for generating extensive qualitative knowledge documents about industrial assets. This approach employs an "entity expansion" technique that systematically discovers and links relevant sensors, assets, and failure modes, thereby enhancing the KG's precision and depth. We provided a mechanism that enables generation of various types of knowledge documents such as $\mathcal{KD}_{\text{Base}}$, $\mathcal{KD}_{\text{Rephrase}}$, $\mathcal{KD}_{\text{QA}}$ and $\mathcal{KD}_{\text{Extend}}$.

- Our experiments demonstrate that the synthetic dataset $\mathcal{KD}_{\text{Base}}$ significantly improved model accuracy from 44.7% to 47.4%, highlighting the effectiveness of diverse data generation. Additionally, the analysis reveals that increasing the number of training tokens yields diminishing returns on performance, emphasizing the need for careful dataset selection and hyperparameter optimization in fine-tuning strategies.

## 2 DOMAIN SPECIFIC DATASET

In this paper, we utilized two separate documents, ISO 14224:2016 ISO (2016) and ISO 17359:2018 CBM (2018), as data source to demonstrate the efficacy of our proposed approach for Industry 4.0 domain. Specifically, we leveraged the former document to generate knowledge for training our model, while the latter document is used to create a benchmark dataset for evaluating the model's performance. ISO is a trustworthy verified piece of information prepared by field experts.

SOURCE DATASET ($\mathcal{D}_{\text{SOURCE}}$)

The ISO 14224:2016 standard provides a comprehensive framework for collecting and exchanging reliability and maintenance data for equipment in the petroleum and gas industries. The standard outlines equipment taxonomy, failure causes, and maintenance actions to enhance reliability, availability, and safety. The knowledge is captured in the form of multiple tabular representations, one for each equipment category, where each row represents an example of a failure mode, and each column corresponds to a piece of equipment that belongs to a particular category.

Table 1: Equipment Failure Data: Structured Knowledge Table

| Equipment Name | Equipment Type | Failure Mode | Failure Examples |
|---|---|---|---|
| Switch-gear | Electrical | External Leakage | Corrosion |
| Pump | Mechanical | Seal Failure | Wear and Tear |
| Valve | Mechanical | Sticking | Contaminants |
| Compressor | Mechanical | Overheating | Insufficient Lubrication |
| Transformer | Electrical | Insulation Breakdown | Aging |

Table 1 is an example of knowledge captured in ISO 14224:2016. Each piece of equipment is assigned a name and type, such as "Switch-gear" being an electrical type of equipment. Furthermore, each piece of equipment can encounter multiple failure modes, with "external leakage" being one example. In total, the dataset comprises 44 assets, and 1795 total failure modes.

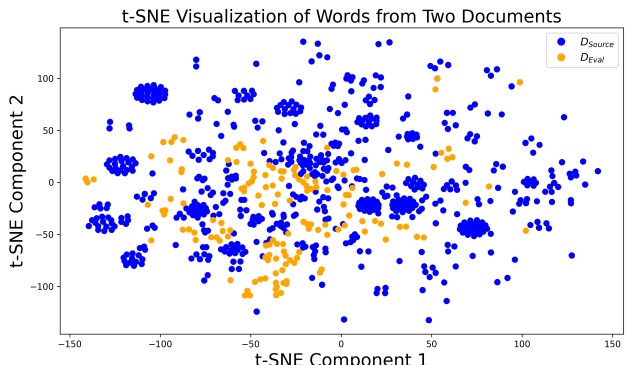

Figure 1: t-SNE Visualization of Word Embeddings from Two Documents

EVALUATION DATASET ($\mathcal{D}_{\text{EVAL}}$)

The ISO 17359:2018 standard outlines the principles of condition-based maintenance for industrial assets. Specifically, this document highlights the use of sensors and parameters to evaluate the condition of an asset across various failure modes. The document is structured in the form of knowledge tables, one for each equipment type, where each row represents a failure mode and each column corresponds to a parameter or sensor that can be used to monitor the asset's condition. The table 2 presents various fault examples related to turbine equipment, indicating how specific sensor readings or parameter changes may occur in the event of a failure. A tick mark (✓) denotes which parameters — Power, Speed, Pressure, Vibration, and Temperature — are affected by each type of fault, providing a clear overview of potential sensor changes associated with these faults.

Table 2: Equipment: Turbine - ✓ indicates that parameter or sensor change if failure occurs

| Failure Mode | Sensor/Parameter Reading | | | | |
|---|---|---|---|---|---|
| | Power | Speed | Pressure | Vibration | Temperature |
| Bearing wear | | ✓ | ✓ | | ✓ |
| Gear Defect | | | ✓ | ✓ | |
| Unbalance | ✓ | | | | ✓ |

We utilize 11 knowledge tables from this document to prepare a multi-choice question-and-answer Industry4-FMSR MCQA dataset. This dataset is used to evaluate the capability of the LLM to answer questions using only its internal knowledge learned during pre-training or supervised fine-tuning. This evaluation dataset allows us to assess the LLM's ability to reason and generate accurate responses based on its acquired knowledge.

Note that, $\mathcal{D}_{\text{source}} \neq \mathcal{D}_{\text{Eval}}$. Figure 1 shows the t-SNE visualization into the semantic relationships between words in the two documents. Any evaluation improvement of a model on $\mathcal{D}_{\text{Eval}}$, where there is no overlap of asset equipments between the $\mathcal{D}_{\text{source}}$ (used for pre-training and finetuning) and $\mathcal{D}_{\text{Eval}}$, strongly suggests that the knowledge gained from the $\mathcal{D}_{\text{source}}$ is transferable to new, unseen data (or "new assets"), and the improvement isn't relying on simple memorization.

## 3 PROBLEM SETTING

Aligning Large Language Models (LLMs) with domain-specific information is a crucial task. Recently, various attempts have been made to generate domain-specific aligned models, such as MediTron-70B for medical domain by Chen et al. (2023), EntiGraph CPT for long passage QA on articles Yang et al. (2024), etc. These approaches typically utilize large-scale corpora with billions of tokens or start with few millions of tokens for generating more synthetic information using teacher model. However, how to accommodate a tiny factual information available in a structured form has not been paid much attention. Our goal is to utilize a limited amount of factual information in a structured format for the Industry 4.0 domain, aiming to uncover the knowledge base of a given LLM surrounding industrial assets for generating more synthetic data.

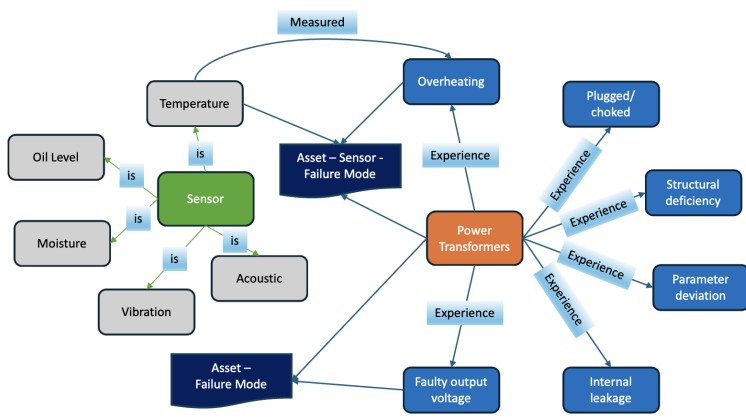

Figure 2: An example of knowledge graph generated around seed entiry using $\mathcal{D}_{\text{source}}$ and our process

We represent structured tabular information, such as Table 1, of $\mathcal{D}_{\text{source}}$ via a knowledge graph (KG), which is a collection of triplets. Formally, a KG is a graph $G = (N, R, E)$, where:

- $N$ is a set of entities such as $equipments$, $failuremodes$, $sensors$, $parameters$, ...
- $R$ is a set of relations such as $monitor$, $experience$, etc, ...
- $E$ is a set of subject-relation-object triplets $(s, r, o)$ where:
    - $s \in N$ (subject entity)
    - $o \in N$ (object entity)
    - $r \in R$ (relation between entities)

The triplets $(s, r, o)$ clearly illustrate how each piece of equipment is linked to its category and failure modes. We begin by assuming that we have extracted a "seed entity" from the source dataset $\mathcal{D}_{\text{source}}$, which will serve as the foundation for generating knowledge documents. While it is conceptually possible to allow the language model (LM) to generate additional seed entities, we argue that expansion based on knowledge knowledge is a more realistic scenario. In this scenario, we are interested in constructing a synthetic knowledge centered around a specific entity.

In addition to equipment, failure modes, and sensors, which are represented as entities in our system, a knowledge document is also an entity that is generated by some procedure and stored within the knowledge graph (KG) as an entity. The creation of knowledge documents involves a multi-step iterative process as outlined in next section. We define the following relations to connect these entities:

- $BelongsTo$: This relation connects equipment to its category
- $Experience$: This relation connects equipment to the failure modes it can encounter
- $Monitor$: This relation connects equipment to the sensor tag for monitoring purpose

Our primary objective is to teach a pretrained language model the knowledge derived from a small set of factual information. However, since the knowledge base is small, the information is highly condensed and lacks diversity in how the underlying knowledge is represented.

## 4 SYNTHETIC DATA GENERATOR: KG + LLM PROMPTING

We employed a two-step iterative approach to generate a large amount of qualitative knowledge documents for industrial assets using a Knowledge Graph (KG) and a Large Language Model (LLM). The approach consists of two phases: knowledge document generation and KG extension.

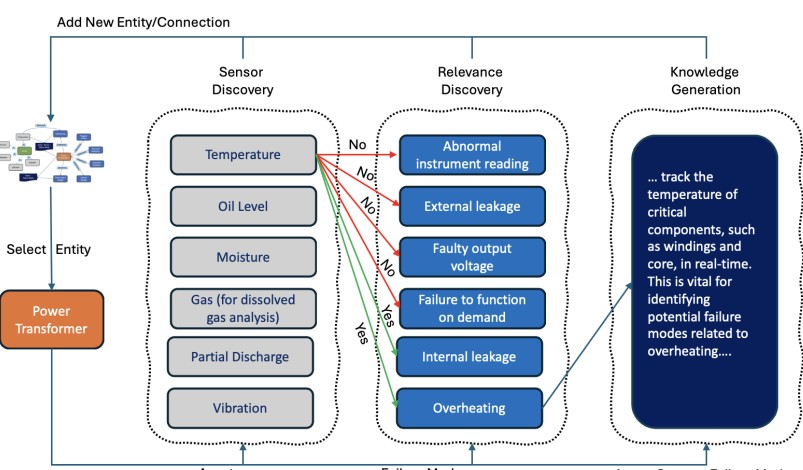

Figure 3: An illustration of full method for generating knowledge document an entity of type equipment

## 4.1 PHASE-I: KNOWLEDGE DOCUMENT GENERATION AND INITIAL KG EXPANSION

The core component of our approach is a focused expansion procedure that takes an entity $e$ of a specific type and extracts all other entities of different types associated with it using the LLM, followed by a relevancy check. This process expands the KG around the entity, enabling us to recursively apply the procedure to further expand the KG. We refer to this process as "entity expansion" and briefly outline in Figure 3.

The entity expansion process consists of three key steps:

- **Sensor Discovery**: Given an entity $e$ of type equipment, we aim to discover all related sensors that can be installed on an asset for its performance monitoring. Specifically, for each $e \in N$, we seek to find a list of entities $O$ that satisfy the relation $(e, \text{'Monitor'}, ?)$. We consider lists since multiple sensors can potentially be used to monitor the asset's performance from different viewpoints. This task will add numerous new nodes to the KG with type sensor.

- **Relevance Generation**: Once sensors are discovered, our next step is to connect sensors and assets to failure modes. Note that not all sensors are useful for detecting all failures, and vice versa. Therefore, we conduct a relevancy check for each pair of sensor and failure mode for a given entity. The list of failure modes are obtained from KG using $(e, \text{'experience'}, ?)$. This relevancy check is crucial for maintaining a high-precision KG. We connect sensors to failure modes using relevancy relations, ensuring that only relevant relationships are established. This task will add numerous links with Yes/No label in KG.

- **Knowledge Generation**: The fine-tuning process of the Large Language Model (LLM) requires passages that capture the relationships between assets, sensors, and failure modes. To address this, this module generates knowledge documents that describe the following three types of relationships:

  - Asset-sensor relationships
  - Asset-failure mode relationships
  - Asset-sensor-failure mode interaction relationships

  LLMs are well-suited for generating summaries or background documents and we can feed single triplet or multiple triplets to generate diverse set of documents. By leveraging this capability, we created a comprehensive set of knowledge documents that provide a detailed understanding of the complex relationships between assets, sensors, and failure modes. The generated documents are also added as entity in KG.

Each steps are achieved via direct prompting the LLM using well-crafted prompt. Table 3 shows detail of each of these step along with the prompt used for LLM query. In summary, we expanded the original KG via various entities of types sensors, and the connection between sensor-failure_mode-asset. Each entity of type sensor unroll a chain of actions in the context of a sub-task.

Table 3: The partial list of sub-tasks in our approach, where for each sub-task we provide its name, a query, a corresponding prompt, and the expected output.

| Sub-task | Query | Prompt |
|---|---|---|
| Sensor Discovery | Power Transformer | What are the sensors that can be installed in the asset `asset_name` for monitoring the performance? Your response should be a numbered list with each sensor name on a new line. For example: 
 `1.  foo` 
 `2.  bar` 
 `3.  baz` |
| Relevance Discovery | Asset, Failure Mode | For the asset `asset_name`, if the failure `failure_mode` occurs, for example, `failure_examples`, can sensor `sensor` help monitor or detect the failure for `asset_name`? Provide the answer in the first line and reason in the second line. |
| Knowledge Generation | Asset, Sensor | You are an expert in industrial asset management, who specializes in failure mode and effects analysis. Given the following input, generate a paragraph of knowledge. Input: `sensor` can be installed in asset `asset_name` for monitoring asset. Knowledge: |

## 4.2 PHASE-II: KNOWLEDGE GRAPH EXTENSION WITH INSTRUCTIONS

The knowledge documents generated so far for the given seed entity in the KG can be utilized to create more documents and/or set of question-answer pairs for instruction tuning. Our unit of operation are triplets that connect the asset with the sensor and failure mode as follows:

$$KD_1 = (e_1, \text{Monitor}, s_1), (e_1, \text{experience}, fm_1), (s_1, \text{relevant}, fm_1)$$

These triplets represent the relationships between the asset ($e_1$), sensor ($s_1$), and failure mode ($fm_1$). Here, $KD_1$ denotes the knowledge document generated for each of the triplet. Let $\mathcal{KD}_{\text{Base}}$ represents all the documents generated so far by following the format as mentioned in $KD_1$. We now explain three approaches we adopted to expand the KG.

### 4.2.1 REPHRASE APPROACH : $\mathcal{KD}_{\text{REPHRASE}}$

We apply a simple data augmentation method to rephrase each $KD_1 \in \mathcal{KD}_{\text{Base}}$. In particular, we adopted a four variation as outlined in  to generate 4 times more documents. These four variation includes such that the generated document look like a toddler will understand (`toddler style`), scholar will understand (`hard style`), Wikipedia style article (`medium style`), and question followed by answer style (`qa style`). All generated documents are stored inside $\mathcal{KD}_{\text{Rephrase}}$. Here is an example prompt used for generating question-answer style of information from a given knowledge document $KD_1$.

> **QA Style Prompt**
>
> A chat between a curious user and an artificial intelligence assistant. The assistant gives helpful, detailed, and polite answers to the questions. Convert the following paragraph into a conversational format with multiple tags of Question: followed by Answer:

### 4.2.2 INSTRUCTION APPROACH : $\mathcal{KD}_{\text{QA}}$

In the second approach, we developed a specialized prompt that outlines a task for a knowledge analyzer to dissect an article concerning an industrial asset. The prompt serves two primary objectives: first, to summarize content as *summary* related to three specified entities: asset name, sensor, and failure mode, ensuring clarity and understanding of their relations. Second, it emphasizes generating thought-provoking questions and corresponding answers that consistently revolve around these three entities, fostering deep analysis and critical thinking. By structuring the response in this format, the prompt encourages exploration of how the sensor data can indicate the potential failure mode of the asset, thereby enhancing comprehension of the asset's operational dynamics and maintenance needs. We plan to generate at least 3 question-answer pairs for each document $KD_1 \in \mathcal{KD}_{\text{Base}}$. Let $\mathcal{KD}_{\text{QA}}$ be a set of 18711 question-answer pairs generated by following this approach. It is important to note that the prompts of $\mathcal{KD}_{\text{QA}}$ has no context information. Therefore, to study whether adding context is helpful to bring more knowledge, we extend $\mathcal{KD}_{\text{QA}}$ with additional context in the prompt which are derived from the generated *summary* in a RAFT manner (Zhang et al. (2024a)). We call it $\mathcal{KD}_{\text{QA-RAFT}}$.

### 4.2.3 KG EXTENSION : $\mathcal{KD}_{\text{EXTEND}}$

In this approach, Knowledge documents are further post processed as a traditional knowledge document to extract the important entity and the relationship between them. The extracted entities are further used to extend the KG by adding more nodes and edge in the graph. In particular, for each knowledge document $KD_1$, the process is consists of three key steps:

- **Entity Extraction** - A set of key entities are extracted and are added into KG
- **Entity Centric Question** - Explain the role of each entity with respect to the $KD_x$
- **Interaction Centric Question** Explore the interaction between pair of extracted entity using $KD_x$

We again followed an entity discovery and relationship explanation procedure.

---

**Prompt 1. Key Entity Extraction Prompt**

As a knowledge analyzer focused on asset management, your task is to dissect and understand an article provided by the user. You are required to perform the following steps:

1. Summarize the Article: Provide a concise summary of the entire article, capturing the main points and themes related to asset management.
2. Extract Entities: Identify and list all significant "nouns" or entities mentioned within the article. These entities should include but are not limited to:

   - Assets: Any equipment or resources referenced in the article.
   - Failure Modes: Specific types of failures or issues that may affect the assets.
   - Sensors: Any monitoring or measurement devices mentioned in relation to asset management.
   - Concepts: Significant abstract ideas or themes central to the discussion of asset management, such as maintenance strategies, risk assessment, or optimization practices.

Try to generate only three to ten key entities. Your response should be structured in a JSON format to organize the information effectively. Ensure that the summary is brief yet comprehensive, and the list of entities is detailed and accurate.
Here is the format you should use for your response:
{ "summary": "", "entities": ["entity1", "entity2", ...] }

---

## 5 EXPERIMENT

We trained using Llama 3.1-8B as the student and Mistral Large 2 as the teacher model. We used 2 NVIDIA A100 80GB GPUs with batch size of 2 per device and gradient accumulation of 1. We used Brain Floating Point to reduce the size of the model parameters and LoRA Hu et al. (2021) to reduce the trainable parameters with rank=8, alpha=32 and dropout=0.1. We used Adam optimizer

Kingma (2014) with Decoupled Weight Decay Regularization Loshchilov (2017) and Cosine Decay Learning Rate scheduler with warmup ratio of 0.05, peak learning rate of $5 * 10^{-6}$ and weight decay factor of 0.01. We used a replay rate of 0.1 of the RedPajama dataset Computer (2023) to minimize the effect of catastrophic forgetting. During training the model processed 3.993 samples/second on average. We also used instruction versions for QA dataset. Due to the highly unstructured format of the generated text, we provide **five question-and-answer** examples as in context learning for a fine tune model which are non-instruction based. The in-context example is not needed for instruction tuned base model.

## 5.1 DATASETS ANALYSIS

In this study, we present a comprehensive analysis of the synthetically generated datasets: $\mathcal{KD}_{\text{Base}}$, $\mathcal{KD}_{\text{Rephrase}}$, $\mathcal{KD}_{\text{QA}}$, and $\mathcal{KD}_{\text{Extend}}$. As mentioned above, we used Mistral Large as the teacher model to generate these datasets. Each dataset comprises of a varying number of passages and question-answer pairs, with totals of 7,307 passages for $\mathcal{KD}_{\text{Base}}$, 23,042 for $\mathcal{KD}_{\text{Rephrase}}$, 17,230 for $\mathcal{KD}_{\text{QA}}$, and an impressive 348,223 for $\mathcal{KD}_{\text{Extend}}$. Utilizing the Llama 3.1 tokenizer, we quantified the total number of tokens for each dataset, resulting in 2.32 million tokens for $\mathcal{KD}_{\text{Base}}$, 7.37 million for $\mathcal{KD}_{\text{Rephrase}}$, 5.4 million for $\mathcal{KD}_{\text{QA}}$, and 334.19 million for $\mathcal{KD}_{\text{Extend}}$. The substantial increase in token count within $\mathcal{KD}_{\text{Extend}}$ signifies its potential utility for fine-tuning large language models, thereby enhancing their performance and adaptability in diverse applications. The accompanying bar plots (Figure 6) visually represent the distribution of passages and tokens across the datasets, highlighting the extensive growth achieved in the later phases of data generation. However, having more data does not necessarily lead to better performance.

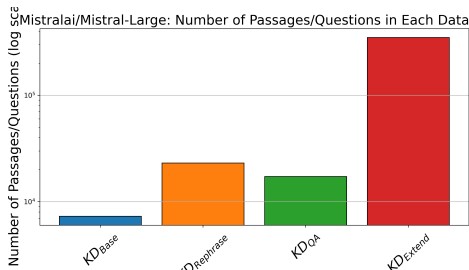

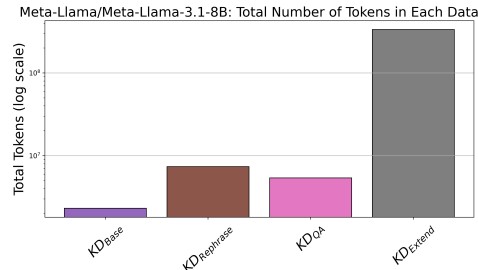

Figure 4: Mistral-Large-2: Number of Passages/Questions

Figure 5: Llama-3.1-8B: Total Number of Tokens

Figure 6: Comparison of Passage Counts and Token Counts Across Datasets

## 5.2 SUMMARY OF FINE TUNING RESULT ON SMALL DATASET

In our experiments, we evaluated the fine-tuning performance of two small datasets, $\mathcal{KD}_{\text{Base}}$ and $\mathcal{KD}_{\text{Rephrase}}$, over three different epochs using the Llama-3.1 Base model, which has a context window of 2048 and an effective batch size of 4. The performance of the base model (without any fine-tuning) is 44.8%, serving as a benchmark for comparison. The results revealed that the accuracy for $\mathcal{KD}_{\text{Base}}$ started at 44.7% but gradually increased to 47.4% by the third epoch, indicating the usefulness of the generated dataset $\mathcal{KD}_{\text{Base}}$. In contrast, $\mathcal{KD}_{\text{Rephrase}}$ exhibited a decrease in accuracy from 46.5% to 45.0% over the same configuration. These findings underscore the observation that simply artificially rephrasing the information is not sufficient to improve model performance. In summary, the synthetic generation of data in $\mathcal{KD}_{\text{Base}}$ achieved better performance, improving by a margin of 3% over 2600+ QA pairs in the test dataset.

## 5.3 SUMMARY OF FINE TUNING RESULT ON LARGE DATASET

Using the results reported in the previous section, we established the performance metrics for the baseline model and the two fine-tuning models, utilizing $\mathcal{KD}_{\text{Base}}$ and $\mathcal{KD}_{\text{Rephrase}}$ at 44.8, 46.2, and 45.4, respectively. We will now explain how we utilized the $\mathcal{KD}_{\text{Extend}}$ dataset.

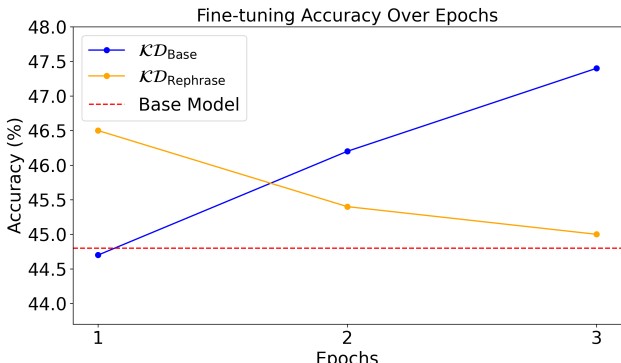

Figure 7: Fine-tuning accuracy for $\mathcal{KD}_{\text{Base}}$ and $\mathcal{KD}_{\text{Rephrase}}$ over three epochs, along with the fixed performance of the base model, demonstrating a decrease in performance as epochs increase.

First, we fixed the epoch at 1 and increased the token size (see Figure 8a) to understand the impact on accuracy. Our analysis demonstrates that increasing the token count did not lead to significant improvements in performance. The plotted line shows fluctuating performance across token sizes, with a peak accuracy of approximately 46.8%. This method exhibits considerable variability but achieves the highest accuracy compared to the other methods. The peaks and troughs suggest that, while there are benefits to increasing the token count, diminishing returns are observed at higher levels. The chart emphasizes the superiority of $\mathcal{KD}_{\text{Extend}}$ in extracting useful representations from the dataset as the token count increases, indicating that this method is particularly effective at leveraging more data. The ability of $\mathcal{KD}_{\text{Extend}}$ to reach higher accuracy levels highlights its potential for applications requiring high performance in fine-tuning tasks.

Next, we used 1000 documents to test the impact of varying epochs. In this case, we fixed the documents and increased the epoch, which also did not show any visible improvement (see Figure 8b). As a result we set the epoch=1 for majority of the experiments.

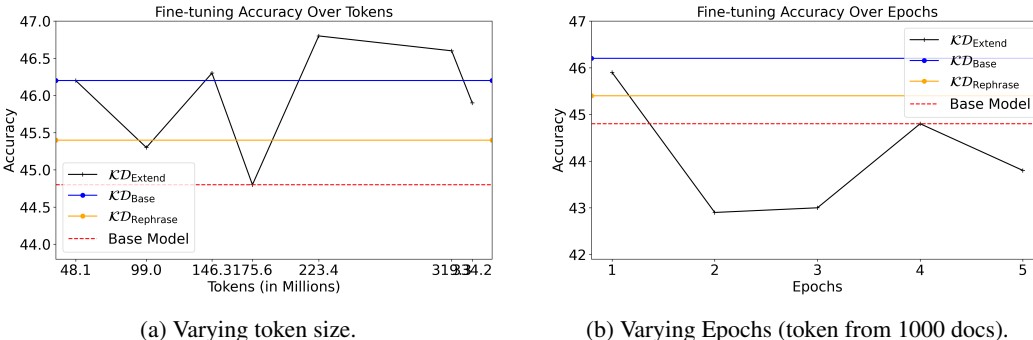

(a) Varying token size.                     (b) Varying Epochs (token from 1000 docs).

Figure 8: Comparison of fine-tuning accuracies across different models.

Figure 9 shows the experimental results demonstrate that the choice of epoch significantly influences the model's accuracy with respect to the number of documents used in training. Epoch 1 provides a more robust performance across a wider range of document counts, indicating a better capacity for learning and generalization. In contrast, Epoch 2 experiences a decline in accuracy with increased training data, highlighting the importance of model training strategies and their implications for performance.

## 5.4 Supervised fine tuning on Instruct model using QA dataset

Following the experiments above, we also conducted fine-tuning on the Instruct model (Meta-Llama-3.1-8B-instruct) to exam whether our approach with $\mathcal{KD}_{\text{QA}}$ and its variation $\mathcal{KD}_{\text{QA-RAFT}}$ could make a difference to the accuracy on $\mathcal{D}_{\text{eval}}$. Notice that since the Instruct model has been specifically

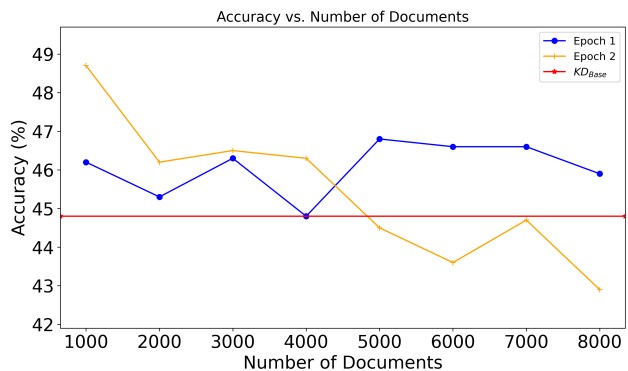

Figure 9: Accuracy vs. Number of Documents: $KD_{Base}$ score is for epoch 1.

trained to follow instructions, the accuracy on a multi-choice question answering task generally outperforms its base non-instruct model. Table 4 has listed the the results of metrics on $\mathcal{D}_{\text{eval}}$.

Table 4: Comparisons of the evaluation metrics on $\mathcal{D}_{\text{eval}}$: Base Instruct vs. Finetuned Instruct model fine-tuned on $\mathcal{KD}_{\text{QA}}$ and $\mathcal{KD}_{\text{QA-RAFT}}$

|  | Accuracy (Exact Match) | Accuracy (excl. Undecided Answers) | Number of Undecided Answers |
| --- | --- | --- | --- |
| Llama-3.1-8B-instruct | 50.1% | 50.7% | 22 |
| Llama-3.1-8B-instruct ft. $\mathcal{KD}_{\text{QA}}$ | 51.3% | 53.0% | 91 |
| Llama-3.1-8B-instruct ft. $\mathcal{KD}_{\text{QA-RAFT}}$ | 45.0% | 48.8% | 242 |

From Table 4, we can see the instruct model fine-tuned on $\mathcal{KD}_{\text{QA}}$ has 1% improvement of exact-match accuracy, and 3% improvement of accuracy if the undecided answers are not counted. The undecided answers are the generations that indicate none of the answers are correct, or hallucinates a non-existing choice. The number of undecided answers is higher after finetuning, mainly because it introduces more noise and bias. This is reflected by the performance result of the RAFT style of supervised fine-tuning deteriorating to 45%. On the other hand, fine-tuning on $\mathcal{KD}_{\text{QA}}$ achieves better balance that avoids too much overfitting but still gains knowledge transfer across domains.

In summary, our idea of generating synthetic data from structured knowledge is validated. Especially $KD_{Base}/KD_{QA}$ being 2.3/5.4 millions synthetic tokens bring close to 4%/3% improvement on the task it has not been trained. Note that, the teacher model has an overall accuracy of 60.5% on our MCQA benchmark which also sets an upper bound on the performance of the student model.

## 6 RELATED WORK

Recent advancements in supervised fine-tuning (SFT) methodologies, such as RAFT Zhang et al. (2024a), emphasize enhancing domain-specific knowledge in models. RAFT systematically trains models using a dataset of question-answer pairs, integrating reasoning chains to generate coherent answers and fostering a more effective retrieval-augmented generation (RAG) framework. Additionally, the synthetic fine-tuning approach advocates for generating more synthetic data Yang et al. (2024). In contrast, TORA Gou et al. (2023) employs a tool-assisted reasoning paradigm where a teacher model guides problem-solving through a systematic method of thinking, acting, and observing. This iterative feedback process refines smaller models and explores various paths to problem resolution, enhancing overall performance. Complementing these, InstructLab Sudalairaj et al. (2024) introduces strategies for improving language model capabilities by eliciting reasoning through brainstorming techniques and generating asset-centric Q&A content. The DDGS framework further integrates diverse textual sources to enhance instruction generation, reflecting the evolving methodologies that emphasize internal reasoning and external information sources in advancing language model performance.

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
