# OpenReview forum: "ALIA: An LLM for Industrial Assets using Synthetic Data"
_ICLR.cc/2025/Conference — Submitted to ICLR 2025_

### Official Review · Reviewer_FDvH · 2024-11-02

**Soundness:** 1
**Presentation:** 2
**Contribution:** 1
**Rating:** 3
**Confidence:** 4

**Summary:**

This paper introduces a method to improve small language models (LLMs) for industrial applications by using synthetic data. The authors focus on creating knowledge documents and question-answer pairs from structured tabular data, which helps fine-tune the LLM for better performance in specific industrial tasks.

**Contributions and Summary:**

1. **Synthetic Data Creation:** The paper presents a way to generate synthetic knowledge documents and question-answer pairs to enhance training data for LLMs. Using knowledge graph and prompt engineering techniques to augment the QA base and so on.

2. **Fine-tuning Approach:** It describes a method for fine-tuning a small LLM (LLAMA 3.1-8B) using the synthetic data, leading to improved model performance.

3. **Experimental Results:** The authors conduct experiments on the Industry4-FMSR MCQA datasets, showing about a 4% performance improvement with the enhanced model compared to the base model.

4. **Knowledge Infusion:** The research highlights the process of extracting structured information from existing documents to effectively integrate domain-specific knowledge into LLMs.

Overall, the paper provides a practical framework for enhancing LLMs in industrial settings through the use of synthetic data.

**Strengths:**

Industrial use case: The methodology outlined in the paper is fairly practical to the industries for specific scenarios with small and complex relations data set that KG might be served to improve the representation efficiency.

Clarity: The paper is generally easy to follow, with a logical flow that makes it easy to follow the authors' arguments and methodologies. Key concepts are explained clearly, and the use of tables and figures helps to illustrate the findings effectively.

**Weaknesses:**

Novelty: The paper presents a industrial practical approach to fine-tuning LLMs specifically for the industrial asset domain. But no/very few novelty of the approach from methodology perspective. The choice of llama or mistral are not well validated. The pros of using KG vs without it is unclear.

Writing issues. However, there are areas where additional clarity could enhance the reader's understanding, particularly in the explanation of the knowledge infusion process, some citations are missing with questionmark, the sections are not organized well so table of contents has some problem such as section 2 is not shown in it.

**Questions:**

Overall, i have a feeling this submission might be proper for industrial track or workshop. I read it with feeling of reading blogs.
Some small questions like:
1. It mentioned one contribution: "Introduce a novel two-step iterative approach that integrates a Knowledge Graph (KG)
with a Large Language Model (LLM) for generating extensive qualitative knowledge documents about industrial assets.". Is the novelty about building a new KG fundamentally or a new continued fine-tuning approach?

2. Why only Llama 3 and Mistral Large? It's better to elaborate a little bit.

3. I failed to find a section like conclusion/limitation so that I'd like to learn what improvements the authors would like to make in the future.

---

### Official Review · Reviewer_qDBh · 2024-11-03

**Soundness:** 2
**Presentation:** 3
**Contribution:** 2
**Rating:** 5
**Confidence:** 4

**Summary:**

This work adapts LLM's by using equipment failure tables from ISO standards. These tables are used to construct a KG, which is subsequently used to generate synthetic documents about the domain of industrial assets. A variety of different document generation strategies are used. A student-teacher setup is used with Llama-3.1-8B-instruct and Mistral 2, and the generated documents are used for fine-tuning.

Evaluation is done via a different set of ISO standards than used in training, and this strategy leads to a small performance increase.

**Strengths:**

- Prompts/instructions provided to LLM's are included
- There is a clear introduction and explanation of the domain
- The paper is well structured and contains detailed explanations making is easy to follow along what has been done.

**Weaknesses:**

- I will start with what I consider the elephant in the room: (intermediate) error analysis. The paper largely focusses on the subsequent performance, but misses the validations at multipe steps of the method. This includes the KG expansion (and 3 subtasks), an evaluation (albeit a sample) of the generated documents and their faithfulness/quality/etc. As the documents are generated in a table-to-text fashion, the most straightforward (and realistic) way would likely be to sample a set and have human experts evaluate them on a bunch of set criteria (e.g. checking whether they are truthful to the KG triples used).
- This brings point 2: motivation and validation of all the extra steps included. The authors include the component of KG expansion. What are the effects (in quantity) for the KG itself? Why all the KD variations, what is the motivation to try them?
- This also follows in the sometimes negative results that are not elaborated upon or concluded. Line 398 "However, having more data does not necessarily lead to better performance. " and Line 449: "Our analysis demonstrates that increasing the token count did not lead to significant improvements in performance". I find these results to be somewhat contradictory compared to general results from the field, but no extra thought is given as to why this might be.
- That also brings me to the results presented and the final conclusion of the strategy. Figure 7, 8 and 9 also show that the fine-tuning strategy also leads to a performance indifferent or worse than the baseline comparison. However the work in general focusses on the small performance increase from the best setting. Why does that specific setting work? Is there a reason for the the others not to work? I consider that these questions stem from a lack of motivation for selecting them in the first place. You can provide a more balanced discussion of the results, both the positive and negative.

This leads to this paper being in a weird place, where it is a work that is relevant, interesting, novel and promising (esspecially for the domain it tries to tackle). But in the way that it is written, it fixates on (likely) work done rather than what are results and contributions that can be used for future work. Some of the missing components will be for questions below, but others can be considered just missing that will require restructuring the paper.

**Questions:**

- What is the quality of documents generated?
- What is the motivation for all the different KG expansion and document generation strategies? What are the differences in results from these strategies? (both quantitative and qualitative)
- What are the takeaways from this paper?
- You state that the number of tokens do not lead to an increase in performance. Do you have an explanation for this?
- The performance increase is there, albeit small. What is still missing from its current state and performance in order to further improve this? (e.g. what is it unable to do?)

---

### Official Review · Reviewer_zrTG · 2024-11-03

**Soundness:** 2
**Presentation:** 1
**Contribution:** 2
**Rating:** 3
**Confidence:** 4

**Summary:**

The authors generate synthetic data using ISO documents for training and evaluating a target LLM (Llama 3.1 8B in this case). By conducting ablation experiments the paper shows that the synthetic data is able to improve the performance of the target LLM on the derived evaluation dataset compared to the baseline target LLM without any domain specific fine tuning. The paper explores multiple ways/approaches to generate the synthetic data and checks the effect on performance once the target LLM is finetuned on the data. The approaches include both summary as well as question answer style synthetic data generation.

**Strengths:**

- The paper uses ISO 14224:2016 ISO (2016) and ISO 17359:2018 CBM (2018) as data sources to generate synthetic train and evaluation set. Such a use case is closer to a real world application of how a company would use LLMs and justifies the case for publishing such datasets which will lead to more novel use of the Language Model technology.
- Synthetically generating large number of tokens need not necessarily improve the model if the net information gain is not much.
- In LORA setup after Epoch 1 model degradation sets in.

**Weaknesses:**

The presentation is lacking in many places with key details missing
- KD_rephrase : The paper lists only the prompt for QA style. Toddler, hard and medium style prompts are not presented. Also what is the intention behind having these 4 variations, do they generate documents with varying amount of knowledge or are they variations with the same knowledge. If its the later case this does not seem to be of much use to an LLM training as there wont be any net new knowledge gain.
- KD_QA : Lines 330 - 331 The claim that the specialized prompt is able to generate thought provoking questions fostering deep analysis and thinking. The authors should present the specialized prompt, baring which its difficult to see how these claims are fulfilled.
- KD_extend: It seems the process is generating large amount of tokens but the net knowledge is not increasing.

Looking at FIg 8(a) for model trained on KD_extend it never beats KD_base which brings into question the efficacy (as claimed in lines 456-457) of generating a large synthetic set when a smaller one can give you better results.

Table 3: Do you check the correctness of the knowledge being generated ? How would the results change if the LLMs are directly finetuned on the ISO docs themselves ?

**Questions:**

- Questions raised in weakness section
- Its not clear whether you will be releasing your datasets to the community ?
- Lines 395-396 How do you ensure there is diversity in these tokens and not just same knowledge repeated multiple times ?
- Line 431: Should 46.2 be 47.4 from line 421 ?
- Line 195: knowledge repeated twice

---

### Official Review · Reviewer_jPrS · 2024-11-04

**Soundness:** 1
**Presentation:** 1
**Contribution:** 1
**Rating:** 3
**Confidence:** 5

**Summary:**

This paper presents an approach for leveraging knowledge stored in a tabular format to generate synthetic knowledge documents and a large set of question-answer pairs within the industrial asset domain, using a Knowledge Graph.

**Strengths:**

The paper effectively explains the need for creating Knowledge Documents to fine-tune and test LLMs.

**Weaknesses:**

Language and Readability: The English used in the paper requires significant improvement. There are numerous typos, and the intended meaning of certain sentences is difficult to discern, which limits the ability to fully understand and evaluate the paper’s contributions. Furthermore, the connections between chapters and the purpose of each section are not consistently described, leaving ample room for enhancement in clarity and coherence.

Figure 2 Explanation: There is no accompanying explanation for Figure 2. The components of the figure need to be clearly described, especially the "Asset-Sensor-Failure Mode" and "Asset-Failure Mode" shapes, which seem to represent processes. Presenting these elements alongside the Knowledge Graph without further clarification appears awkward.

Entity Expansion Process: It is stated that "Sensor Discovery" and "Relevance Discovery" are conducted for Entity Expansion, but it remains unclear whether this process is exclusively performed through prompts input into the LLM. Does this mean the entire ISO Document is provided as input to query the LLM? It is difficult to understand why this process is necessary given that the Knowledge Graph has already been created. More detailed explanation is needed.

Generated Knowledge in Figure 3: Upon reviewing the generated knowledge in Figure 3, it appears challenging for the LLM to generate this form of output using only the triple relations from the previously built KG. Additional explanation on how this is achieved would be helpful.

Examples of Knowledge Documents (KDs): Please provide specific examples for each type of Knowledge Document (KD) to clarify the contribution.

**Questions:**

In the experimental section starting from Section 5.2, what does “accuracy” specifically refer to? More concrete details on how accuracy was measured through experimentation are necessary. While the KG Extension process is understandable, it is difficult to grasp how the question-answer set was created or tested due to the lack of explanation on these points.

---

### Meta-Review · Area_Chair_frAQ · 2024-12-19

**Metareview:**

**Summary**


This paper aims to propose a model to leverage tabular data for generating syntatic data to train LLMs. The domain is specific industrial applications. The learning dataset seems to consists of question-answering pairs derived from these tabular data. The authors report an increase of performance on a dataset called Industry4-FMSR MCQA datasets.

**Strengths**

- The paper carefully prompts/instructions provided to the LLMs.
- The paper uses ISO standars to generate synthetic data


**Weaknesses**


- The method in the paper is composed of multiple steps that are not correctly evaluated. The focus is only on the presentation of the final performance gain.

- Results of the paper seems to confirm an emerging properties of LLMs, that is, LLMs are able to memorize. This affect the capabilities of a correct evaluation as described in the notion of data contamination [1] and as happened for LMs [2].


**Final remarks**

The paper has not been defended by the authors during the discussion phase and, thus, no answer to the questions of the reviewers have been provided.

**References**

* [1] Inbal Magar, Roy Schwartz, Data Contamination: From Memorization to Exploitation, 2022
* [2] Leonardo Ranaldi, Elena Sofia Ruzzetti, and Fabio Massimo Zanzotto. 2023. PreCog: Exploring the Relation between Memorization and Performance in Pre-trained Language Models.

**Additional Comments On Reviewer Discussion:**

No discussion has been initiated by the authors.

---

### Decision · Program_Chairs · 2025-01-22

Reject